# Anaplastic Thyroid Carcinoma Histologically Mimicking a Plasmacytoma

**DOI:** 10.3390/diagnostics10010029

**Published:** 2020-01-08

**Authors:** Joshua H. Mo, Donald Tan, Xiaohui S. Zhao, Tjoson Tjoa, Beverly Y. Wang

**Affiliations:** 1Department of Pathology and Laboratory Medicine, UC Irvine School of Medicine, Irvine, CA 92617, USA; zhaox@uci.edu (X.S.Z.); bevwang@uci.edu (B.Y.W.); 2Department of Otolaryngology-Head and Neck Surgery, University of California, Irvine, CA 92697, USA; donaldt2@uci.edu (D.T.); tjoat@uci.edu (T.T.)

**Keywords:** anaplastic thyroid carcinoma, plasmacytoma, mimic

## Abstract

Anaplastic (undifferentiated) thyroid carcinoma (ATC) is a rare malignancy which may arise from transformation of a pre-existing differentiated carcinoma. We report the unique case where a lesion of thyroid origin presented with the histological features of mature plasma cells. Immunohistochemistry confirmed the lesion to be an anaplastic thyroid carcinoma arising from papillary thyroid carcinoma. A tumor mimicking a malignancy of a different cellular origin can lead clinicians to incorrect treatment approaches. Careful correlation with clinical details and knowledge of these unique presentations is important for reaching the correct diagnosis.

## 1. Introduction

Anaplastic (undifferentiated) thyroid carcinoma (ATC) is a rare malignancy that accounts for an estimated 1–2% of thyroid cancers [1]. ATCs are highly aggressive solid tumors, with a median survival of less than 6 months after diagnosis [2,3,4,5]. They typically occur in patients who are 65 years of age or older [1].

A large body of literature has explored the mutational landscape underlying ATCs. Through whole-exome sequencing, one group observed that these malignancies have variations gathered within the MAPK, Ras, and ErbB signal transduction pathways; with recurrent mutations in *TP53*, *BRAF*, and *RAS*-family of genes. All the ATC lines had a combination of *BRAF* and *TP53*, or *NRAS* and *TP53* [6]. For anaplastic carcinomas and poorly differentiated carcinomas, activating mutations in *BRAF* were exclusive to tumors arising from papillary thyroid carcinomas [7].

There is no current standardized treatment for ATC, and interventions are mostly aimed to improve local control [8,9]. Rarely, the patient’s tumor may be meaningfully resected without extensive spread at the time of diagnosis. These patients can undergo a multimodal curative intent treatment consisting of surgical resection combined with chemoradiation [10]. However, most patients present with disease that has invaded nearby structures and metastasized. Palliative intent treatment often consists of neoadjuvant and adjuvant chemotherapy along with surgical resection to prevent airway compromise [11]. Novel, molecular targeted therapies are used in ongoing clinical trials. These are targeted to the mutational landscape of the tumor and include, but are not limited to, B-RAF inhibitors, mTOR inhibitors, and multikinase inhibitors [8,12,13,14].

ATC is unusual compared to other thyroid malignancies in that it is typically diagnosed based on clinical symptoms, rather than fine-needle aspiration (FNA) on a suspicious thyroid nodule. These symptoms include rapidly enlarging neck mass, dyspnea, dysphagia, neck pain, Horner’s syndrome, stroke, and hoarseness due to vocal cord paralysis [15].

There are three main histological growth patterns to ATC: spindle cell, pleomorphic giant cell, and squamoid. Most tumors demonstrate one or more of these histological patterns. Rare histological variants also exist, such as the paucicellular variant and the rhabdoid variant. The only variant of ATC with known prognostic significance is the paucicellular variant, which in some studies was found to affect younger patients and have a more indolent course [16]. There are various reports in the literature of ATC mimicking other entities such as Riedel’s thyroiditis, squamous cell carcinoma, and even a benign histiocytic proliferation [17,18,19]. We present a unique variant of ATC that demonstrates a plasmacytic morphology.

## 2. Case Presentation

A 54-year-old woman was referred to our tertiary care Head and Neck Surgery clinic for evaluation of a rapidly enlarging left neck mass. She reported the neck mass had developed over the past four months, caused mild discomfort, and had associated unintentional weight-loss. The patient endorsed a remote 20+ year smoking history but denied any family history of head and neck malignancies. Physical exam during this visit revealed an eight-centimeter nodal conglomerate in the left supraclavicular area and diffusely enlarged thyroid gland with a roughly 5 cm right thyroid mass. She had previously undergone fine needle aspiration (FNA) of the neck mass at an outside facility which demonstrated rare degenerated atypical cells suspicious for malignancy, and a positron emission tomography scan (PET) scan with fluorodeoxyglucose (FDG) avid lesions in the neck and lungs concerning for metastases, which also showed tracheal deviation secondary to mass effect. Inhouse computed tomography (CT) of her neck showed a necrotic mass, while CT angiography of the chest showed lung lesions consistent with her previous imaging (Figure 1). Repeat FNA cytology of the supraclavicular mass revealed a poorly differentiated carcinoma. These cells were positive for TTF-1 and PAX8 by immunohistochemistry, and the treatment team continued to suspect a thyroid origin. After consultation with the oncology team, the surgery team and the patient elected for surgical resection of the left neck mass for diagnostic purposes, as well as thyroid surgery for palliation, given the mass effect and tracheal deviation. With diagnostic surgery as the next step, additional thyroid lesion workup such as a thyroid ultrasound, or measurement of her serologic thyroglobulin, anti-thyroglobulin, and calcitonin were not taken at this time. The patient then underwent a left neck dissection, and her left thyroid lobe and isthmus were resected. Additionally, the level 4 supraclavicular mass with multiple surrounding firm, enlarged nodes was resected en bloc. The right thyroid lobe and left central neck compartment contents were left in place to minimize surgical morbidity. The left thyroid lobe and supraclavicular mass were sent for pathology.

The histological sections showed a dense, monotonous mass of cells with abundant amphophilic cytoplasm and open nuclei that contained coarse chromatin and prominent perinuclear hof. A large fraction of the cells was binucleated (Figure 2). This was a peculiar finding as these features are consistent with mature plasma cells. There were very few cells with plasmablastic or anaplastic features, and a distinct lack of pleomorphism or prominent nucleoli. The large lymph node that was taken showed the same cellular architecture, with almost complete erasure of the native lymphoid tissue (Figure 2). These histologic findings on the initial H&E were peculiar and further immunohistochemical stains verified the diagnosis. The lesion was positive for cytokeratin AE1/AE3, TTF-1, and PAX-8 with a markedly increased Ki-67 proliferative index at over 50%, consistent with an aggressive carcinoma of thyroid origin. Medullary thyroid carcinoma can have a similar cellular morphology, however, stains for Calcitonin were negative, which suggests against this entity. A negative p63 stain helped rule out a primary squamous cell carcinoma, while a negative SOX10 stain helped rule out a neuroendocrine or melanocytic origin. The workup for any lymphoid or histiocytic origin turned up negative, as the tumor cells did not stain for CD3, CD20, CD45, CD79a, CD138, or MUM-1. Additionally, there were areas of papillary thyroid carcinoma in both specimens collected (Figure 2). Further genetic testing found that the lesion contained a *BRAF* (V600E) mutation. With these findings alongside the patient’s clinical presentation, we were able to conclude that the patient developed an aggressive anaplastic thyroid carcinoma with a plasmacytic morphology from her background papillary thyroid carcinoma. The patient and family elected to undergo palliative radiation therapy and cytotoxic chemotherapy, as well as pursue further molecular profiling and B-RAF inhibitor treatment in the future.

## 3. Discussion

The diagnosis of anaplastic thyroid carcinoma is heavily clinically based, and histologic diagnosis can be challenging and confusing. Treatment intervention and outcomes heavily differ between the various thyroid carcinomas as well as plasma cell dyscrasias. In this case, the continued suspicion for anaplastic thyroid carcinoma was confirmed by the immunohistochemical results and the *BRAF* mutation that was found. The plasmacytoid morphology we observed is a unique histological manifestation of ATC. While there are reports of ATC mimicking other entities, as we previously described, this is the first case of a plasmacytoid mimicking ATC to our knowledge. This finding did not change the diagnosis or management of the patient. However, it should be noted that many diagnoses are made off cytology from fine needle aspiration. The paucity of cells acquired may not be sufficient for confirmatory immunohistochemistry. The finding of these cellular features alone could mislead clinicians to pursue a workup toward a plasma cell dyscrasia. Knowledge of these unique presentations, as well as apt communication between pathologists and clinicians ensure patients receive the correct treatments in these types of cases.

## Figures and Tables

**Figure 1 diagnostics-10-00029-f001:**
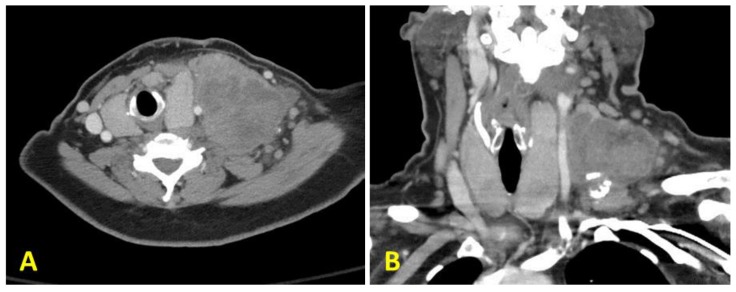
Computed tomography (CT) scan of the patient’s neck showing a large mass located at the level 4, with central cystic necrosis ((**A**): cross section and (**B**): coronal section). Axial section (**C**) and coronal section (**D**) taken from a CT angiogram of the chest showing metastatic lung lesions.

**Figure 2 diagnostics-10-00029-f002:**
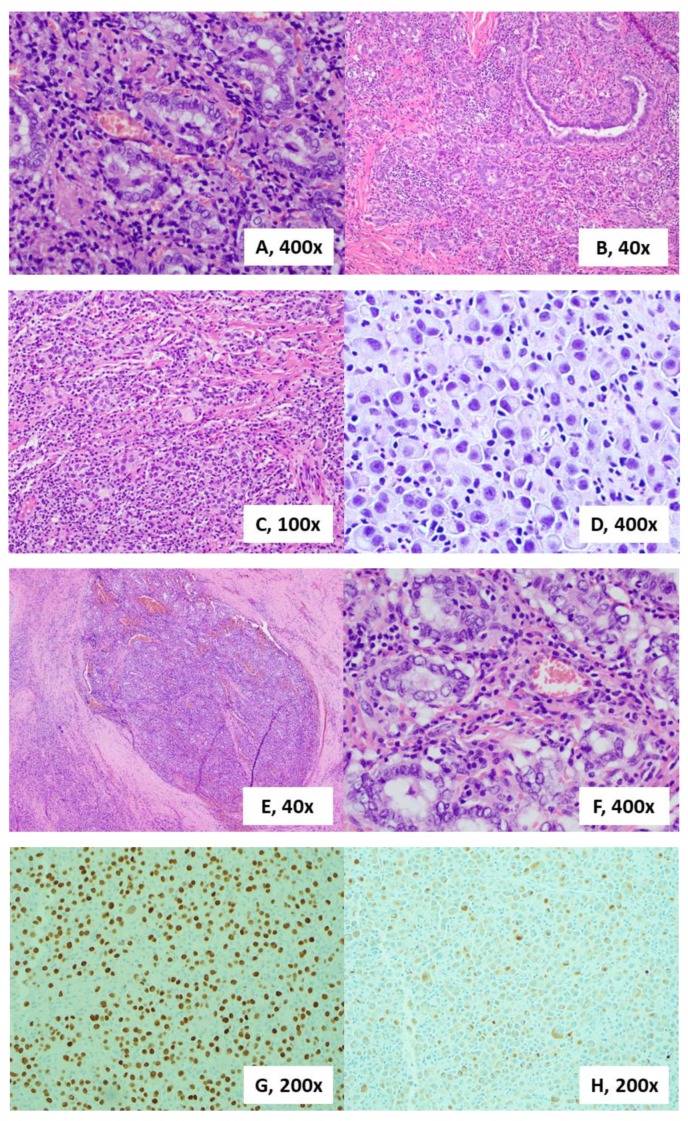
Histologic sections of the left thyroid lobe showing areas of papillary thyroid carcinoma (**A**), H&E 400×; with a transitional morphology to microfollicular, solid sheet and plasmacytic pattern of growth (**B**), H&E 40×; at a lower magnification showing sheets of plasmacytic cellularity (**C**), H&E 100×; and at a higher magnification showing binucleated cells with mature plasma cell features (**D**), H&E 400×. Histologic sections of the supraclavicular mass showing a lymph node with metastatic papillary thyroid carcinoma (**E**), H&E 40×; and a higher magnification showing a solid, undifferentiated component (**F**), H&E 400×. Positive immunohistochemical staining for TTF-1, 200× (**G**) and PAX-8, 200× (**H**) on sections of the left thyroid lobe.

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
