# Peer review of "Anaplastic Thyroid Carcinoma Histologically Mimicking a Plasmacytoma"

_diagnostics, 2020, doi:10.3390/diagnostics10010029_

Round 1

Reviewer 1 Report

In the manuscript submitted by Mo et al. Anaplastic Transformation of Papillary Thyroid Carcinoma Mimicking Plasmacytoma histologically diagnosed ATP is presented. The diagnosis of ATP is routine clinically and FNA biopsy-based. However, limitations could arise in the detection of ATP due to the small amount of sample tissue. The object is important because ATP grows very rapidly and the histopathologic correlation with clinical symptoms is valuable for reaching a rapid diagnosis and the accurate patient treatment options.

The manuscript is well-written and the results are clearly presented.

I suggest authors to add the microphotographs regarding TTF-1 and PAX8 positivity in FNA diagnosed, poorly differentiated carcinoma (line 47). Similar to CT/PET/CT, these are the manuscript „starting points” for histopathological tissue analysis. Please, describe each part of multipart figures with lettered panel label, for example, 1A, etc. (line 46, line 60). Please, add scale bars/magnifications in the figure legends. In the figure 2D, the first arrow shows incorrect leukocyte. Please point correctly both arrows.

Author Response

I suggest authors to add the microphotographs regarding TTF-1 and PAX8 positivity in FNA diagnosed, poorly differentiated carcinoma (line 47). Similar to CT/PET/CT, these are the manuscript „starting points” for histopathological tissue analysis. Please, describe each part of multipart figures with lettered panel label, for example, 1A, etc. (line 46, line 60). Please, add scale bars/magnifications in the figure legends. In the figure 2D, the first arrow shows incorrect leukocyte. Please point correctly both arrows.    -micrographs of TTF-1 and PAX8 were added -figure legends were redone to include all information -magnifications were added to the panels -arrows were removed

Reviewer 2 Report

It is an interesting case, mainly due to the unusual presentation of a large extra thyroid mass. The case has very serious bias.

the title should be refined. There is no support for the diagnosis of plasmocytoma, except some histological data: "there were very few cells with plasmablastic or anaplastic features". In the next sentence the authors refer to the "large lymph node with the almost complete erasure of the native  lymphoid tissue". This is confusing: the authors consider  that the case was  an exceptional rare (<50 cases reported) of lymph node plasmocytoma or an extra medullary plasmocytoma? there are no consistent facts to link the mass with plamoscytoma; there are no serologic determination and the imaging tests are not conclusive in this direction. there is a complete lack of specific investigation for thyroid: thyroid ultrasound, serologic thyroglobulin and anti-thyroglobulin, calcitonin. The thyroid ultrasound would exclude the thyroid nodule; there is no sentence referring to the structure of thyroid; this should be the particularity of the case report: the cervical mass of anaplastic thyroid cancer and NO primary tumor in the gland. Is there any nodule in the gland? The CT suggests in both lobes in the lower poles some hypodensities, that needed to be confirmed in ultrasound. we have been lost between many immunohistochemical staining and we know just few data about the standard evaluation of patient with thyroid cancer.   there are serious confusions in the text regarding figure 1; this is described in the manuscript as showing a lung nodule and the legend is presented as liver and lymph node metastasis Figure 1 C - left image is an axial section of upper abdomen CT, NOT an PET/CT; the right image is not a fused PET/CT image, it is a PET image; there is NO fused PET/CT image in figure 1. Moreover the arrow shows a lung nodule in the right lung, not a lymph node as is mentioned in the text of the figure's legend and the liver has no radiopharmaceutical pathologic uptake. in the discussion section the authors mentioned that the patient went "complete surgery" line 85, while in lines 47-48-49 is mentioned that there was a partial thyroidectomy and diagnostic excision of the mass.

Author Response

the title should be refined.

-title was changed

There is no support for the diagnosis of plasmocytoma, except some histological data: "there were very few cells with plasmablastic or anaplastic features". In the next sentence the authors refer to the "large lymph node with the almost complete erasure of the native  lymphoid tissue". This is confusing: the authors consider  that the case was  an exceptional rare (<50 cases reported) of lymph node plasmocytoma or an extra medullary plasmocytoma? there are no consistent facts to link the mass with plamoscytoma; there are no serologic determination and the imaging tests are not conclusive in this direction.

-areas of the paper were rewritten to clearly define the diagnosis of ATC and the histological presentation of a plasmacytic morphology 

there is a complete lack of specific investigation for thyroid: thyroid ultrasound, serologic thyroglobulin and anti-thyroglobulin, calcitonin. The thyroid ultrasound would exclude the thyroid nodule; there is no sentence referring to the structure of thyroid; this should be the particularity of the case report: the cervical mass of anaplastic thyroid cancer and NO primary tumor in the gland. Is there any nodule in the gland? The CT suggests in both lobes in the lower poles some hypodensities, that needed to be confirmed in ultrasound. we have been lost between many immunohistochemical staining and we know just few data about the standard evaluation of patient with thyroid cancer.

-explanation added as to why the specific investigations for a thyroid lesion was not taken in lieu of a diagnostic surgery. additional information regarding the structure of the thyroid was added, including the presence of a nodule.

  there are serious confusions in the text regarding figure 1; this is described in the manuscript as showing a lung nodule and the legend is presented as liver and lymph node metastasis Figure 1 C - left image is an axial section of upper abdomen CT, NOT an PET/CT; the right image is not a fused PET/CT image, it is a PET image; there is NO fused PET/CT image in figure 1. Moreover the arrow shows a lung nodule in the right lung, not a lymph node as is mentioned in the text of the figure's legend and the liver has no radiopharmaceutical pathologic uptake.

-Figure 1 was redone to now have the lung metastases, and the figure legend was corrected.

in the discussion section the authors mentioned that the patient went "complete surgery" line 85, while in lines 47-48-49 is mentioned that there was a partial thyroidectomy and diagnostic excision of the mass

-discussion section was rewritten to include the correct information

Reviewer 3 Report

This case report by Mo and co-workers is an interesting description of an aggressive thyroid carcinoma with plasmacytic features, a histological phenomenon which in the endocrine sphere is mostly associated to medullary thyroid carcinoma. The case is rather unique, and the manuscript is for most parts well written and illustrated, but some clarifications are needed - as well as more extensive coverage of the literature.

1. The authors should clearly define the diagnosis. In the abstract, the tumor is entitled "poorly differentiated carcinoma" ( = PDTC). Do the authors mean "undifferentiated thyroid carcinoma (= ATC) ? The same goes for the cytology report (row 46), was the pre-operative diagnosis consistent with PDTC, Bethesda VI? To me, PDTC for most parts is an histological diagnosis using either the Turin or Memorial Sloan Kettering criteria. Please comment on this, and make it clear to the readers what type of tumor that was ultimately diagnosed.

2. The Introduction is a bit too short, especially the part detailing the underlying genetics. TP53 mutations are not uniquely found in ATCs, but also seen in PDTCs and subsets of aggressive well differentiated thyroid cancers - see for example PMID: 26878173 and PMID:31621921. The underlying mutational landscape as a whole has been previously published as well in PMID: 25576899 and PMID: 26980298, to name a few.

3. I think the authors at least need to discuss potential differential diagnoses, is there a possibility of this lesion being either a calcitonin-negative MTC with plasmacytic features alternatively a primary squamous cell carcinoma? I assume the authors did not stain for calcitonin gene related peptide (CGRP) or sequenced the RET gene from a clinical standpoint? Were there amyloid deposits in the stroma? And for SCC, did the authors consider CK5, p40 or p63 IHC? Please elaborate in the manuscript, even thought the clinical history and the co-occurrence with a PTC makes these differential diagnoses unlikely.

4. Is this the first case of plasmacytoid ATC ever to be reported? Have the authors checked for other case reports regarding this? I think the manuscript could contain a passage of text in which this aspect is discussed.

Author Response

The authors should clearly define the diagnosis. In the abstract, the tumor is entitled "poorly differentiated carcinoma" ( = PDTC). Do the authors mean "undifferentiated thyroid carcinoma (= ATC) ? The same goes for the cytology report (row 46), was the pre-operative diagnosis consistent with PDTC, Bethesda VI? To me, PDTC for most parts is an histological diagnosis using either the Turin or Memorial Sloan Kettering criteria. Please comment on this, and make it clear to the readers what type of tumor that was ultimately diagnosed.

-The abstract was corrected to include the correct description. The cytology report is not on a thyroid FNA, but instead on the supraclavicular mass. This diagnosis would not be under the Bethesda classification (for thyroid FNA - PDTC, etc.)

2. The Introduction is a bit too short, especially the part detailing the underlying genetics. TP53 mutations are not uniquely found in ATCs, but also seen in PDTCs and subsets of aggressive well differentiated thyroid cancers - see for example PMID: 26878173 and PMID:31621921. The underlying mutational landscape as a whole has been previously published as well in PMID: 25576899 and PMID: 26980298, to name a few.

-The introduction was rewritten and an additional section on treatment was added. The mutational landscape section was updated to include the most recent data and areas were incorporated to the treatment section.

3. I think the authors at least need to discuss potential differential diagnoses, is there a possibility of this lesion being either a calcitonin-negative MTC with plasmacytic features alternatively a primary squamous cell carcinoma? I assume the authors did not stain for calcitonin gene related peptide (CGRP) or sequenced the RET gene from a clinical standpoint? Were there amyloid deposits in the stroma? And for SCC, did the authors consider CK5, p40 or p63 IHC? Please elaborate in the manuscript, even thought the clinical history and the co-occurrence with a PTC makes these differential diagnoses unlikely.

-The case was updated to include additional immunohistochemistry and genetic sequencing. While a calcitonin-negative MTC and primary SCC are possibilities, the IHC findings combined with a BRAF V600E mutation in the context of the clinical presentation solidified the diagnosis as ATC. The clinicians did not pursue any other differentials and this is detailed in the updated report.

4. Is this the first case of plasmacytoid ATC ever to be reported? Have the authors checked for other case reports regarding this? I think the manuscript could contain a passage of text in which this aspect is discussed.

-The discussion section was rewritten regarding this. From our literature search, this is the first presentation of plasmacytoid ATC